# Creatine Alleviates Doxorubicin-Induced Liver Damage by Inhibiting Liver Fibrosis, Inflammation, Oxidative Stress, and Cellular Senescence

**DOI:** 10.3390/nu13010041

**Published:** 2020-12-24

**Authors:** Nouf Aljobaily, Michael J. Viereckl, David S. Hydock, Hend Aljobaily, Tsung-Yen Wu, Raquel Busekrus, Brandon Jones, Jammie Alberson, Yuyan Han

**Affiliations:** 1School of Biological Sciences, University of Northern Colorado, Greeley, CO 80639, USA; nouf.aljobaily@unco.edu (N.A.); michael.viereckl@unco.edu (M.J.V.); Albe4122@bears.unco.edu (J.A.); 2School of Sport and Exercise Science, University of Northern Colorado, Greeley, CO 80639, USA; david.hydock@unco.edu (D.S.H.); raquel.busekrus@unco.edu (R.B.); brandon.jones@unco.edu (B.J.); 3Independent Statistician, Fort Collins, CO 80525, USA; hend.aljobaily@hotmail.com; 4Obstetrics and Gynecology Department, University of Washington, Seattle, WA 98115, USA; wuyenyen40@gmail.com

**Keywords:** liver damage, doxorubicin, creatine, oxidative stress, senescence, liver fibrosis, inflammation

## Abstract

**Background**: Treatment with the chemotherapy drug doxorubicin (DOX) may lead to toxicities that affect non-cancer cells including the liver. Supplementing the diet with creatine (Cr) has been suggested as a potential intervention to minimize DOX-induced side effects, but its effect in alleviating DOX-induced hepatoxicity is currently unknown. Therefore, we aimed to examine the effects of Cr supplementation on DOX-induced liver damage. **Methods:** Male Sprague-Dawley rats were fed a diet supplemented with 2% Cr for four weeks, 4% Cr for one week followed by 2% Cr for three more weeks, or control diet for four weeks. Animals then received either a bolus i.p. injection of DOX (15 mg/kg) or saline as a placebo. Animals were then sacrificed five days-post injection and markers of hepatoxicity were analyzed using the liver-to-body weight ratio, aspartate transaminase (AST)-to- alanine aminotransferase (ALT) ratio, alkaline phosphatase (ALP), lipemia, and T-Bilirubin. In addition, hematoxylin and eosin (H&E) staining, Picro-Sirius Red staining, and immunofluorescence staining for CD45, 8-OHdG, and β-galactosidase were performed to evaluate liver morphology, fibrosis, inflammation, oxidative stress, and cellular senescence, respectively. The mRNA levels for biomarkers of liver fibrosis, inflammation, oxidative stress, and senescence-related genes were measured in liver tissues. Chromosomal stability was evaluated using global DNA methylation ELISA. **Results:** The ALT/AST ratio and liver to body weight ratio tended to increase in the DOX group, and Cr supplementation tended to attenuate this increase. Furthermore, elevated levels of liver fibrosis, inflammation, oxidative stress, and senescence were observed with DOX treatment, and Cr supplementation prior to DOX treatment ameliorated this hepatoxicity. Moreover, DOX treatment resulted in chromosomal instability (i.e., altered DNA methylation profile), and Cr supplementation showed a tendency to restore chromosomal stability with DOX treatment. **Conclusion:** The data suggest that Cr protected against DOX-induced hepatotoxicity by attenuating fibrosis, inflammation, oxidative stress, and senescence.

## 1. Introduction

Doxorubicin (Adriamycin, DOX) is an effective chemotherapeutic agent used to treat various cancers such as breast cancer, bladder cancer, stomach cancer and leukemia [1]; however, severe side effects including nausea, vomiting, extravasation, severe cardiotoxicity, and hepatoxicity have been reported. In addition, drug resistance has limited its application in cancer patients [1,2,3]. The molecular mechanisms underlying the anti-cancer effects of DOX are multifaceted and are not completely understood, although it is well known that DOX can enter the nucleus of the cell and interfere with DNA replication by inhibiting topoisomerase II [4]. Furthermore, DOX undergoes redox cycling leading to the generation of reactive oxygen species (ROS) [1,5] and damage to cellular lipids, proteins, and DNA, which leads to cell death [6]. Doxorubicin treatment has also been found to promote systemic inflammation by damaging the small intestine and releasing microbial endotoxins into circulation, thus stimulating pro-inflammatory pathways and enhancing hepatic inflammation [7]. It has also been suggested that DOX upregulates genes involved in cell senescence, which may also contribute to DOX’s anticancer effect [8]. The multitude of cytotoxic effects makes DOX effective across various human cancers [9], but DOX’s toxicity in healthy, non-cancer cells and tissues limits its use in combating cancer in patients [10,11].

One such tissue negatively affected by DOX is the liver. Liver toxicity is a significant clinical consideration with DOX treatment as elevated liver enzymes have been observed in up to 40% of patients receiving DOX [12]. Moreover, DOX-induced ROS in the liver, specifically superoxide anion, compromises mitochondrial function [13]. The exact mechanisms behind DOX-induced hepatoxicity, however, are not completely understood. Therefore, the first aim of this study was to explore mechanisms behind DOX-induced liver damage.

Because DOX-related side effects limit its use as an anticancer agent, interventions to minimize its toxicity have received increased attention. Supplementing the diet with creatine (Cr) is beginning to show promise. Creatine is a naturally produced compound used primarily in the phosphocreatine system to provide energy in the brain, heart, and skeletal muscle, and its use as a dietary supplement is generally considered safe [14]. Additionally, Cr may function as an antioxidant capable of reducing oxidative stress [15], and consequently, it been suggested as a treatment for oxidative stress-induced cardiovascular damage [16]. Previous studies have shown that Cr has protective effects against DOX in skeletal myofibers and cardiomyocytes [17]. Supplementing the diet with Cr increases intracellular reserves of phosphocreatine in skeletal myofibers and cardiomyocytes [18]. Moreover, Cr supplementation has been shown to reduce the skeletal myofibers dysfunction and fatigue that accompanies DOX treatment [19]. Although studies have shown Cr’s ability to attenuate DOX toxicity in skeletal myofibers and cardiomyocytes, the role that Cr plays in managing DOX hepatoxicity is currently unknown. Therefore, the present study also aimed to evaluate the effects of dietary Cr supplementation on DOX-induced hepatotoxicity by evaluating several indices including liver fibrosis, inflammation, oxidative stress, senescence, cellular damage, and chromosome stability.

## 2. Materials and Methods

### 2.1. Reagents, Chemicals, and Primers

All reagents were purchased from VWR (Radnor, PA, USA) and Thermo Fisher Scientific (Waltham, MA, USA) unless otherwise indicated. Rats were obtained from Envigo (Indianapolis, IN, USA). Doxorubicin hydrochloride was purchased from Wedgewood Pharmacy (Swedesboro, NJ, USA), and all primers were purchased from Invitrogen (Carlsbad, CA, USA).

### 2.2. Animals and Treatment

All procedures were approved by the Institutional Animal Care and Use Committee at the University of Northern Colorado (protocol number 1711CE-DH-R-20 approved on 28 December 2018) and were in accordance with the Animal Welfare Act guidelines. Male Sprague-Dawley rats (*n* = 60) were randomly assigned to one of three groups: standard rodent chow feeding (control, CTRL, *n* = 20); chow supplemented with 2% Cr for 4 weeks (2% Cr, *n* = 20); or chow supplemented with 4% Cr for 1 week followed by 2% Cr for 3 more weeks (4%/2% Cr, *n* = 20). After 4 weeks of feeding, rats were further subdivided to receive either 15 mg/kg DOX i.p. (doxorubicin hydrochloride, Wedgewood Pharmacy, CTRL/DOX, *n* = 10; 2% Cr/DOX, *n* = 10; 4%/2% Cr/DOX, *n* = 10) or 0.9% sterile saline i.p. as a placebo (CTRL/SAL, *n* = 10; 2% Cr/SAL, *n* = 10; 4%/2% Cr/SAL, *n* = 10). Five days following the injections, the animals were euthanized with sodium pentobarbital. Liver tissue was then collected immediately post sacrifice, weighed, embedded in Optimal Cutting Temperature (O.C.T.) fluid, snap-frozen in liquid nitrogen and stored at −80 °C. Liver tissue not embedded in O.C.T. was also collected and snap frozen in liquid nitrogen for later analysis.

### 2.3. Blood Collection and Peripheral Serum Chemistry

Approximately 5 mL of peripheral blood was collected from each rat in the tubes, and was centrifuged at 1000 rpm for 5 min to separate the blood. Serum was then collected from the top fraction and stored at −80 °C. In order to assess the overall liver function, serum samples were sent to the Colorado State University Veterinary Teaching Hospital Diagnostic Laboratory to evaluate the levels of aspartate transaminase (AST), alanine aminotransferase (ALT), and the AST to ALT ratio, alkaline phosphatase (ALP), lipemia, and total bilirubin.

### 2.4. Tissue Preparation and Hepatic Histology

Hematoxylin and eosin (H&E) staining procedures were employed for the general tissue morphology. Briefly, 8 µm frozen liver sections were fixed in 10% neutralized buffered formalin (NBF) for 10 min, and then washed in 95% ethanol. Slides were then washed three times in tap water and stained with hematoxylin for 40 s followed by three tap water washes. Sectioned slides were then dipped four times in ammonia water and rinsed four times with tap water followed by 20 dips in 95% ethanol. Then the sections were stained in eosin for 10 s. Slides were dehydrated twice in 95% ethanol, three times in 100% ethanol, treated twice with xylene and mounted using mounting media. Images were then taken at 20× magnification.

To evaluate liver fibrosis, Picro-Sirius red stain was used. Briefly, 8 µm frozen liver sections were air-dried for 3 min and placed in xylene for 10 min. Samples were then rehydrated in 100%, 90% and 70% ethanol, respectively. Nuclei were stained with hematoxylin (VWR, CO) for 30 s, and then washed in running water for 10 min. Slides were then stained in Picro-Sirius red stain for 60 min and washed two times in 0.5% acidified deionized water. Slides were then dehydrated in 70%, 90% and 100% ethanol and dipped in xylene twice and mounted with mounting media. Images were obtained at 4× magnification. Three biological replicates from each group were used with fields randomly chosen for quantification. The collagen deposition in selected samples was quantified using ImageJ software [20].

Senescence was evaluated with a commercially available kit (OZ Biosciences, CA) according to the manufacturer’s protocol. Tissues were fixed with fixing solution for 10 min and then rinsed three times in phosphate-buffered saline (PBS). Slides were then immersed in SA-beta-gal solution (pH 6.0) overnight at 37 °C in the dark and were then fixed in 4% paraformaldehyde for 10 min followed by three washes in PBS, counterstained with nuclear fast red for 20–30 s and rinsed in running water for 2 min. Slides were then dehydrated in 95% ethanol and 100% ethanol followed by three treatments of xylene and mounted using mounting media. Images were then obtained at 10× magnification.

### 2.5. Gene Expression

Total RNA was extracted using a PureLink™ RNA Mini Kit (Invitrogen, Carlsbad, MA, USA) and the manufacturer’s protocol. RNA was then treated with DNase I (Thermo Scientific, Waltham, CO, USA) and converted to cDNA using an Applied Biosystem High-Capacity cDNA Reverse Transcription Kit (Applied Biosystem, Carlsbad, CA, USA). Quantitative reverse transcription polymerase chain reaction (qRT-PCR) was then carried out to obtain mRNA expression levels (Appendix A). All genes were then normalized to β-Actin (ACTB), and all data presented have a sample sizes of *n* = 4 unless otherwise indicated.

### 2.6. Immunofluorescence Staining

Frozen tissue sections were fixed in 10% NBF, washed three times with PBS and incubated in 10% goat serum blocking buffer for 20 min, which was followed by three PBS washes. Samples were then incubated overnight in the primary antibodies (CD-45 from Biolegend, CA, 8-hydroxy-2′-deoxyguanosine (8-OHdG) from Santa Cruz, Port Aransas, TX, USA) at 4 °C and washed three times with PBS. Sections were then incubated in the secondary antibody for 45 min and washed three times with PBS. Slides were mounted with mounting media containing DAPI (Electron Microscopy Sciences, West Orange, PA, USA), and the secondary antibody was also applied to slides without primary antibody incubation as the negative control. All images were obtained at 20× magnification and resolved using confocal microscopy.

### 2.7. Global DNA Methylation

A MethylFlash Methylated DNA 5-mC Quantification Kit (Epigentek) was used to analyze global DNA methylation levels. Genomic DNA was isolated from snap-frozen tissue using a DNAzol kit according to the manufacturer’s protocol (Thermo Scientific, CO, USA). A standard curve, 2 μL of samples, positive controls, and negative controls were plated in the kit’s 96-well plate. Solutions were then mixed and incubated at 37 °C for 60 min followed by binding solution being removed from all wells. Wells were then washed three times using diluted wash buffer. Next, 50 µL of 5-mC detection complex solution was added to each well and incubated at room temperature for 50 min. The 5-mC detection complex solution was decanted, and the wells were washed five times with wash buffer. At this point, detection solution was added to induce a color change, and the reaction was stopped when the 5% positive control turned dark blue. Absorbency was then analyzed at 450 nm with the relative percentage of methylation calculated in reference to the CTRL/SAL group.

### 2.8. Statistical Analyses

All data are reported as mean ± standard error (SE) unless otherwise indicated and were checked for normality using the Shapiro-Wilk test. ANOVA and non-parametric Kruskal-Wallis tests were performed using the R program to determine significance between group differences. When a significant difference between groups was observed, the F-value was observed, Tukey’s and Dunn’s post hoc tests were performed to identify where differences existed. Significance was set at *p* < 0.05.

## 3. Results

### 3.1. Creatine Supplementation Tended to Improve Doxorubicin-Induced Liver Damage

We proposed that Cr could alleviate liver damage caused by DOX (Figure 1A). To determine this, we first assessed overall liver health by evaluating serum AST, ALT, ALP, lipemia, and total bilirubin (Appendix A) as well as the liver-to-body weight ratio and AST-to-ALT ratio. We found a trend in elevated liver-to-body weight ratio in the CTRL/DOX group compared to CTRL/SAL, and a significant decrease in liver-to-body weight ratio was observed in the 4%/2% Cr/SAL and 2% Cr/DOX groups compared to the CTRL/DOX group (*p* < 0.001) (Figure 1B upper panel). Furthermore, CTRL/DOX showed a trend toward a higher AST-to-ALT ratio than CTRL/SAL and a trend toward a decreased AST-to-ALT ratio in 4%/2% Cr/DOX was observed (Figure 1B, lower panel). We then evaluated liver histological changes using H&E staining, and interestingly, white blood cell infiltration was noted in the periportal region of the liver when rats were given Cr supplementation alone specifically in the 4%/2% Cr/SAL group; however, no significant histological changes were observed in the CTRL/DOX group when compared to CTRL/SAL (Figure 1C).

### 3.2. 2% Creatine Treatment Reduced Doxorubin-Induced Liver Fibrosis

Liver fibrosis usually occurs when chronic injury or inflammation leads to a build-up of scar tissue, which can be indicated by increased collagen deposition [21]. After serum chemistry revealed liver damage, we assessed collagen deposition in frozen liver tissues using Picro-Sirius red staining. A significant difference in fibrosis was noticed between groups (Figure 2A,B) [20] with an increasing trend in fibrosis in CTRL/DOX when compared to CTRL/SAL. The 2% Cr/DOX group showed the lowest level of fibrosis (less than 5%) (Figure 2B), and a significant decrease in fibrosis was observed in both 2% Cr/DOX and 4%/2% Cr/DOX groups when compared to CTRL/DOX (*p* < 0.01). We further investigated fibrotic-related biomarkers, fibronectin-1 (FN-1), which shows enhanced expression during fibrogenesis [22]. The expression of FN-1 was enhanced in all three DOX treatment groups, the expression of FN-1 in the 4/2%Cr/Dox group tends to decrease when compared to the CTRL/DOX group (Figure 2C).

### 3.3. Pro-Inflammation Markers Induced by Doxorubicin Are Attenuated by Creatine Supplementation

Hepatic inflammation is part of the hepatic repair process that is triggered by liver injury caused by different etiology such as fat accumulation, drug induction and viral infection. After observing potential white blood cell infiltration in the liver with H&E staining (Figure 1C), we investigated the likelihood of DOX inducing inflammation and whether Cr supplementation could alleviate it. An increase in CD45 positive cells was observed in the CTRL/DOX group. CD45 Expression decreased in the 2% Cr/DOX and 4%/2% Cr/DOX groups when compared with CTRL/DOX group (Figure 3A). However, there was no evidence of CD45 positive cells in the 2% Cr/SAL and 4%/2% Cr/SAL groups. The positive staining of CD45 prompted us to investigate which inflammatory factors might play a role. Previous reports have shown that the nuclear transcription factor NF-κB plays a central role in this process [23], and IL-1β has previously been shown to act as a pro-inflammatory cytokine in chronic liver injury [24]. Consistent with the CD45 staining, NF-κB expression was elevated in the CTRL/DOX group (*p* < 0.05); however, no significant decrease in mRNA level was detected in the 2% Cr/DOX or 4%/2% Cr/DOX groups (Figure 3B). Further, CTRL/DOX showed elevation at the transcription level of pro-inflammatory cytokine, IL-1β, when compared to CTRL/SAL, with a trend towards reduced levels of gene expression in both 2% Cr/DOX and 4%/2% Cr/DOX (Figure 3C). Besides, we noticed the mRNA expression of NF-κB and IL-1β increased in 4%/2% Cr/SAL compared with CTRL/SAL, which is not consistent with CD-45 staining, suggesting the potential inflammatory stress when supplementing with a higher dosage of Cr per se. 

### 3.4. Creatine Supplementation Alleviated Doxorubicin-Induced Oxidative Stress

Oxidative stress is caused by the overproduction of ROS and/or the inability to quench ROS. This has previously been observed in rats receiving DOX and has been suggested to be part of the etiology for hepatotoxicity [13]. Studies have shown that DOX induces oxidative stress, which leads to inflammation [25]; thus, it was vital to investigate whether the inflammatory response observed could be due to oxidative stress. The oxidative stress biomarker 8-OHdG was increased in the CTRL/DOX group when compared to CTRL/SAL, and this was decreased in the 4%/2% Cr/DOX group but not in the 2% Cr/DOX group (Figure 4A). Similarly, elevated levels of nuclear factor erythroid 2 (NRF-2) were observed in CTRL/DOX (*p* < 0.01) and 2% Cr/DOX when compared with the CTRL/SAL group (*p* < 0.01), and decreased levels of NRF-2 were shown in 4%/2% Cr/DOX when compared with the CTRL/DOX group (Figure 4B). Like pro-inflammation biomarkers, we also observed enhanced expression of NRF-2 in 4%/2% Cr/SAL, which was not consistent with protein levels of 8-OHdG when compared with CTRL/SAL, indicating a higher dosage of Cr increases the risk of oxidative stress in healthy rats. Taken together, our findings suggest that DOX does increase oxidative stress whereas 4%/2%Cr supplementation prior to DOX treatment alleviates oxidative stress.

### 3.5. Higher Dose of Creatine Showed More Alleviation in the Senescence Caused by Doxorubicin

Cellular senescence is defined as cell-cycle arrest and cells are incapable of proliferation. These phenomena mainly occur due to genetic changes, yet more recently, it was found that they could be induced by external factors such as radiation and drug treatment [26]. Furthermore, several studies have shown that cellular senescence increases when cells are treated with DOX [27,28]. Thus, we assessed cellular senescence using β-galactosidase (an enzyme only present in senescent cells) staining and qRT-PCR analysis for monocyte chemoattractant protein-1 (MCP-1) expression (a chemokine known to be associated with inducing cellular senescence [29].). Our data suggest that when rats received no DOX (i.e., SAL groups), there was no indication of cellular senescence whereas all groups that received DOX showed cellular senescence (yellow arrows). Both 2%Cr/DOX and 4%/2%Cr/DOX showed decreased senescence when compared to CTRL/DOX (Figure 5A). Additionally, MCP-1 mRNA expression was increased in CTRL/DOX when compared to CTRL/SAL, and 2% Cr/DOX and 4%/2% Cr/DOX showed a decrease of MCP-1 mRNA expression when compared to CTRL/DOX (Figure 5B), which is consistent with senescence staining. Collectively, these data indicate that DOX treatment promoted liver senescence that were engaged in increased transcription of MCP-1, but Cr supplementation attenuated this observed cell senescence.

### 3.6. Chromosomal Instability with Doxorubicin Treatment Which Was Rescued with Creatine Feeding

A previous study suggested that DOX can alter gene expression patterns by epigenetic modification in the heart [30]. Further, global hypomethylation has been found to cause chromosomal instability in various cancers [31,32]. Hence, it was important to explore the effects of DOX treatment on global DNA methylation status in the liver and determine the role Cr supplementation plays in this process. Significant hypomethylation was observed in CTRL/DOX compared to the CTRL/SAL group (*p* < 0.01), and although not significant, the 2% Cr/DOX and 4%/2% Cr/DOX groups showed a trend towards attenuation of hypomethylation (*p* = 0.06 vs. CTRL/DOX) (Figure 6). Creatine supplementation could potentially rescue the chromosomal instability caused by DOX by increasing genomic DNA methylation; however, more work is needed in this area.

## 4. Discussion

This study reports DOX-induced liver damage by observable increases in the ALT-to-AST ratio. Furthermore, elevated liver fibrosis, inflammation, oxidative stress, and senescence and decreased global methylation were observed following DOX treatment. Rats fed with Cr before DOX administration, however, showed reduced liver damage following DOX treatment as demonstrated by a decreased liver to body weight ratio, and decreased levels of inflammation, fibrosis, senescence, and oxidative stress. The current study demonstrates the potential therapeutic effect of Cr in alleviating DOX-induced liver damage.

Many studies have shown the cytoprotective effects of Cr in cardiac myocytes and skeletal myocytes in rat models [16,17,19,33]. The current study found similar benefits of Cr in the liver. Cr reduced DOX-induced liver damage by alleviating the fibrosis, inflammation, oxidative stress and senescence. Elevated serum AST-to-ALT is a commonly used metric for detecting hepatotoxicity [34], and we found elevated AST-to-ALT levels in both CTRL/DOX and 2% Cr/DOX groups, which indicates liver stress. 4%/2% Cr/Dox, on the other hand, had AST-to-ALT ratios with values comparable to the SAL-treated groups.

This study applied two different Cr feeding regimens to investigate potential dose-dependent effects. 4%/2% Cr/DOX showed decreased AST-to-ALT ratios, liver fibrosis biomarker expression, liver inflammation, oxidative stress when compared with the 2% Cr/DOX group, suggesting that a one-week Cr loading phase (i.e., high Cr intake) may be a better treatment option for enhancing liver function and decreasing the liver inflammation associated with DOX treatment. No differences, however, were observed in terms of senescence or methylation between the two different Cr feeding regimens. Interestingly, the 4%/2% Cr/SAL group presented a moderate increase in mRNA expression of the liver oxidative stress biomarker NRF-2 and the inflammation biomarkers NF-κB and IL-1β along with observable lymphocytic penetration. Taken together, the data indicate that the higher dosage of Cr (4%/2%) was, more affective at alleviating DOX-induced liver damage than the lower Cr dose (2% Cr). The higher dose of Cr, however, could potentially increase inflammation and oxidative stress in the healthy rat liver.

Our study showed that IL-1β is the primary pro-inflammatory factor upregulated in DOX-induced liver inflammation, which was reduced with Cr feeding. Previous reports have also shown that DOX-induced oxidative stress could cause inflammatory cells to release cytokines such as TNF-α, IL-1β and IL-6 [35,36]. Furthermore, NF-κB expression was elevated with DOX treatment regardless of whether animals were fed Cr or not. This suggests that Cr may alleviate DOX-induced inflammation independent of the NF-kB pathway, which has also been shown in a model of skeletal muscle inflammation [37].

Cellular senescence is a protective mechanism to protect potentially damaged cells against mutation; however, when cellular senescence becomes chronic, it causes the tissue to be more susceptible to damage [38,39]. Senescent cells are usually arrested in the G1/S phase and express P16 and P21 cell cycle inhibitors [40]. Indeed, our data suggest enhanced senescence in DOX-induced liver damage and reduced senescence with Cr supplemented groups receiving DOX, which was consistent with MCP-1 gene expression. Cellular senescence plays a crucial role in chronic injury, and studies have shown senescence of hepatocytes and cholangiocytes in chronic hepatitis B and C [41,42], alcoholic liver damage [43], primary sclerosing cholangitis (PSC) [43,44], and non-alcoholic fatty liver diseases [45]. The mechanism underlying hepatic senescence in chronic liver disease has been found to be stress related, and previous studies have shown that oxidative stress could cause DNA damage and subsequently induce cellular senescence [46,47,48]. Indeed, we found oxidative stress-related gene expression showed a similar trend as the change in senescence level; however, more work is needed to investigate whether the effects of Cr on cellular senescence is directly or indirectly related to the reduction of oxidative stress.

The present study did not observe a dramatic change in liver fibrosis in DOX groups, which seems paradoxical to other reports. This could be explained by the fact that studies have used various treatment frequencies and concentrations. Another possibility is that the one-time, bolus injection of DOX only causes acute liver damage, and fibrosis occurs at a later stage of liver damage. Actually, DOX toxicity is dose-dependent with clinicians limiting cumulative doses to 450 mg/m^2^ to prevent cardiotoxicity [49].

Furthermore, it has been observed that global DNA hypomethylation is associated with liver damage. Studies have shown that DOX can cause global hypomethylation by decreasing the level of methylation enzymes [50,51], and similarly, we have demonstrated hypomethylation in DOX treated groups in the current study. Others have shown that global hypomethylation could cause chromosomal instability and make chromosomes more vulnerable to damage [32]. Hypomethylation is also present in an early-stage liver fibrosis model [52], but it is unclear whether DNA methylation status is related to any specific senescence, fibrosis, or inflammation genes.

In summary, this study reports the protective effects of Cr in alleviating DOX-induced liver damage in rats. Supplementing the diet with Cr was effective in reducing the fibrosis, inflammation, oxidative stress and senescence that accompany DOX treatment. The results of this study suggest that Cr administration could be a potential adjuvant therapeutic approach to attenuate DOX-induced liver damage. Investigating the long-term effects of Cr is still needed to further assess its safety in cancer patients undergoing chemotherapy, and further research is warranted to explore the safety and efficacy of Cr supplementation as an intervention to protect against DOX-induced hepatotoxicity.

## 5. Conclusions

DOX administration caused the increase in liver to body weight ratio, liver fibrosis, liver inflammation, oxidative stress, senescence and genomic instability. In contrast, supplementation with Cr decreased DOX-induced liver damage as demonstrated by reduced liver to body weight ratio, liver fibrosis, and liver inflammation. Furthermore, the higher dosage of creatine induced a potential risk of inflammation stress in healthy rats. 

## Figures and Tables

**Figure 1 nutrients-13-00041-f001:**
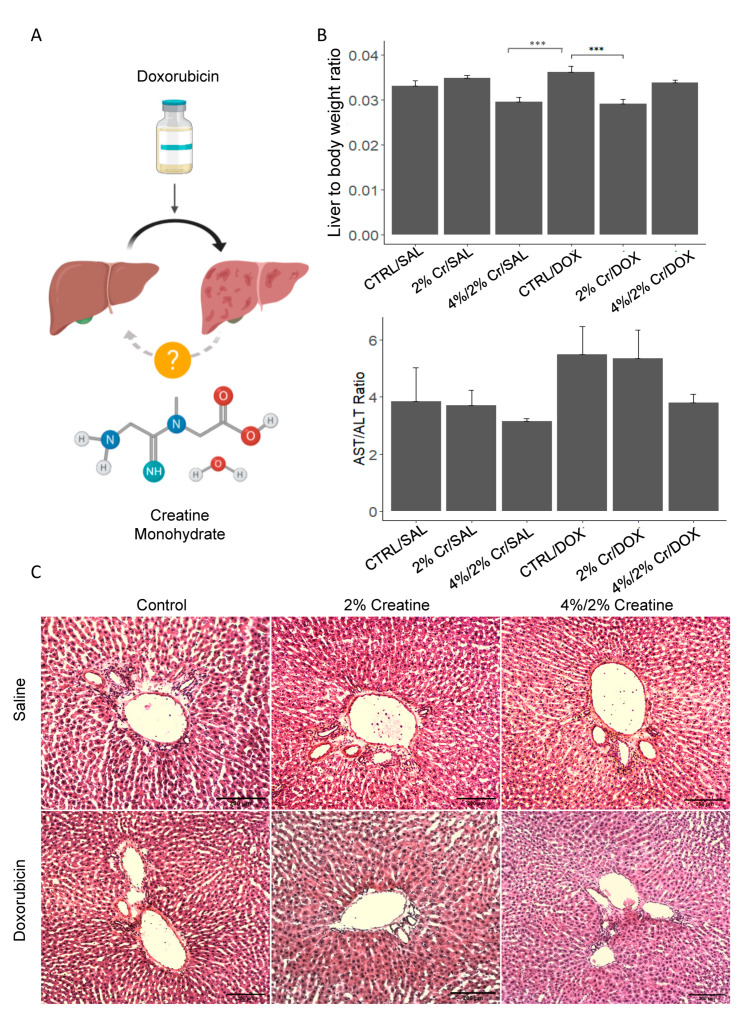
Evaluation of the effect of doxorubicin (DOX) on the overall health of liver and the role of creatine (Cr) in reducing the damage. Schematic illustration describing the hypothesis of the present study (**A**). There was a significant decrease in the liver-to-body weight ratio when 2% Cr/DOX and 4%/2%Cr/SAL were compared toCTRL/DOX ((**B**), top row). The AST-to-ALT ratio tended to be elevated in CTRL/DOX and showed a trend toward a reduction in 4%/2% Cr/DOX ((**B**), bottom row). Immune cell infiltration was observed in the representative H&E images, but no major change was noted among the groups. Pictures were taken under the magnification of 20×. (**C**). Data is reported as mean ± SE, *** *p* < 0.001.

**Figure 2 nutrients-13-00041-f002:**
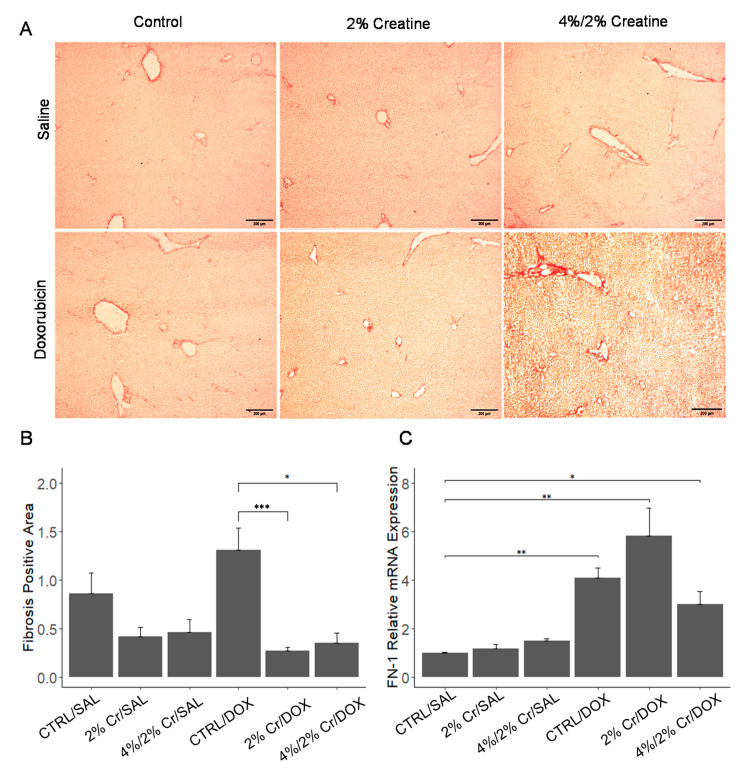
Liver fibrosis was measured by Picro-Sirius Red staining and mRNA expression of the fibrosis biomarker FN-1. Representative Sirius Red staining showing mild fibrosis in CTRL/SAL group (**A**). When images were quantified, a tendency of higher fibrosis was observed in CTRL/DOX when compared to the CTRL/SAL. Further, there was a significant decrease in 2% Cr/DOX and 4%/2% Cr/DOX when compared to CTRL/DOX (**B**). mRNA expression of FN-1 was elevated in CTRL/DOX and tend to reduce when DOX was combined with 4%/2% Cr (**C**). Three biological samples were imaged at 10× magnification and quantified using image J software. Four biological replicates are used, and mRNA expression is relative to β-Actin (ACTB). Data is reported as mean ± SE. * *p* < 0.05, ** *p* < 0.01, *** *p* < 0.001.

**Figure 3 nutrients-13-00041-f003:**
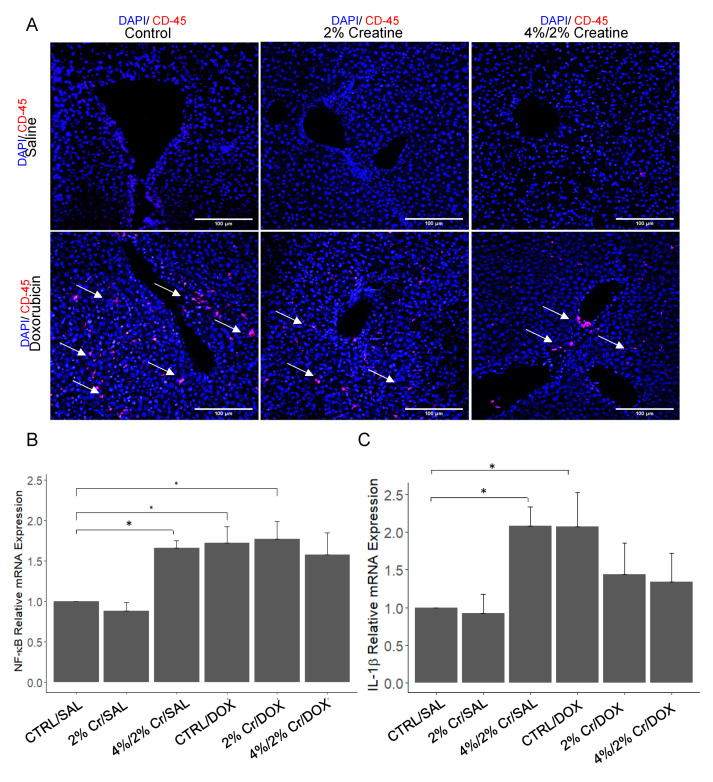
Immunofluorescence staining and mRNA expression of inflammation-related genes. Antibody against CD45 is shown in red and nuclei stained with DAPI are shown in blue. Arrows illustrate positive staining of CD45. (**A**). Positive immunofluorescence staining for CD45 in DOX treated group (bottom row) but not in the control groups (top row) (**A**). mRNA expression of NF-κB were elevated in CTRL/DOX when compared to CTRL/SAL and no significant decrease was detected in the 2% Cr/DOX or 4%/2% Cr/DOX groups when compared to CTRL/DOX (**B**). mRNA expression of IL-1β was elevated in CTRL/DOX and 4%/2% Cr/SAL when compared to CTRL/SAL, and a trend toward a reduction in mRNA level was detected in the 2% Cr/DOX or 4%/2% Cr/DOX groups when compared to CTRL/DOX (**C**). The image was obtained under 20× magnification. Four biological replicates are used in Figure 3B and six biological replicates are used in Figure 3C, and mRNA expression is relative to ACTB. Data is reported as mean ± SE. * *p* < 0.05.

**Figure 4 nutrients-13-00041-f004:**
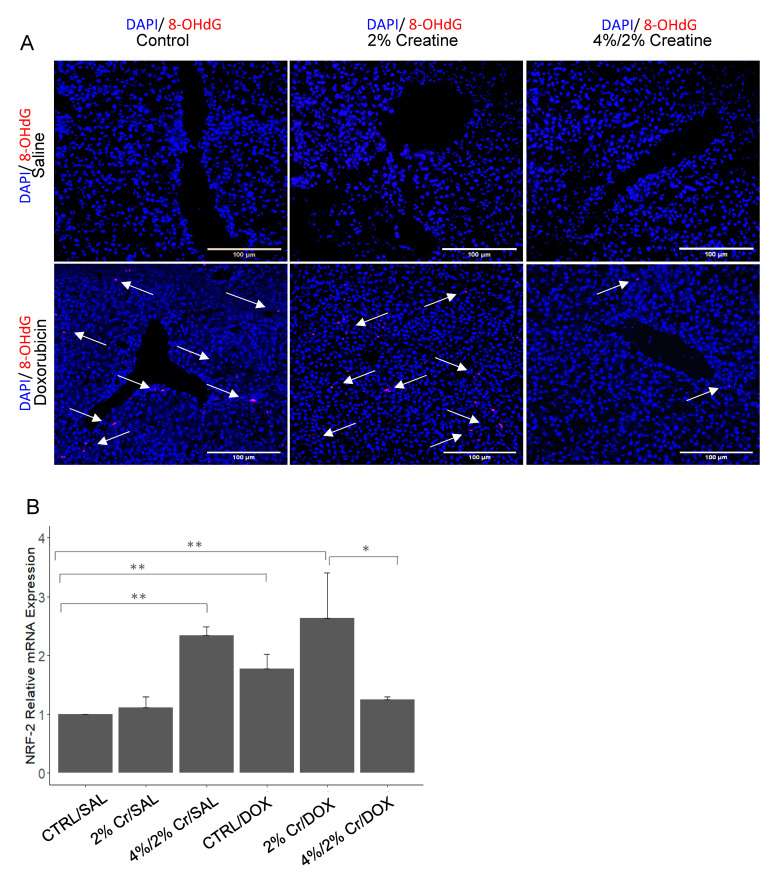
Immunofluorescence staining and mRNA expression of oxidative stress-related genes. Antibody against 8-OHdG is shown in red and the nucleus stained with DAPI is shown in blue. The arrow shows the positive staining of 8-OHdG. 8-OHdG was expressed in DOX treated groups (bottom row), while there was no indication of 8-OHdG expression in controls (top row). Pictures were taken under 20× magnification (**A**). Low mRNA expression of NRF-2 was observed in CTRL/SAL and 2% Cr/SAL while elevated mRNA expression of NRF-2 was detected in 4%/2% Cr/SAL, CTRL/DOX and 2% Cr/DOX (**B**). The image was obtained under 20× magnification. Four biological replicates are used, and mRNA expression is relative to ACTB. Data is reported as mean ± SE. * *p* < 0.05, ** *p* < 0.01.

**Figure 5 nutrients-13-00041-f005:**
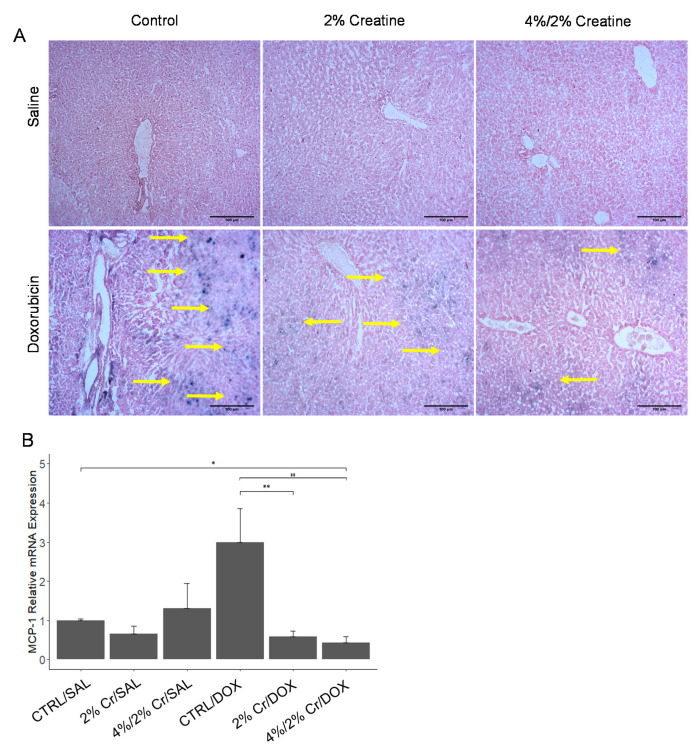
Liver senescence was measured using β-galactosidase staining and qRT-PCR. Histologically, no senescence was observed in control groups (top row), while more senescence (bottom row) was observed in groups that received DOX treatment. Both 2%Cr/DOX and 4%/2%Cr/DOX showed decreased senescence when compared to CTRL/DOX. The blue spot indicates the senescence sites (yellow arrow) (**A**). Images were obtained under 20× magnification. mRNA expression of MCP-1 was highest in CTRL/DOX but decreased in groups receiving Cr supplementation when compared to CTRL/DOX (**B**). Six biological replicates are used, and mRNA expression is relative to ACTB. Data is reported as mean ± SE. * *p* < 0.05, ** *p* < 0.01.

**Figure 6 nutrients-13-00041-f006:**
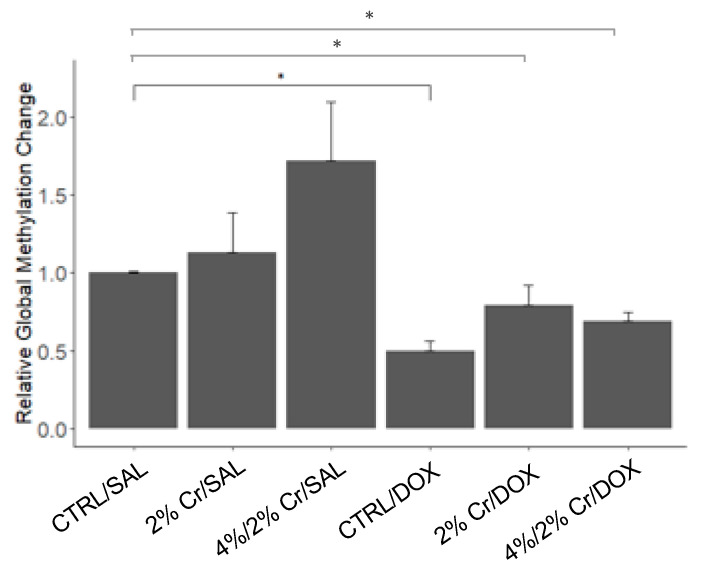
Quantification of 5-methylcytosine (%5-mC) to determine the change in global methylation and thus chromosomal stability using 5-mC ELISA. Higher levels of global methylation were found in groups that did not receive DOX whereas lower levels of global methylation were detected in groups that received DOX treatment. Additionally, when DOX was combined with Cr supplementation, levels of global methylation tended to increase. Data are reported as mean ± SE. * *p* < 0.05.

## Data Availability

The data presented in this study are available on request from the corresponding author.

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
