# Peer review of "Creatine Alleviates Doxorubicin-Induced Liver Damage by Inhibiting Liver Fibrosis, Inflammation, Oxidative Stress, and Cellular Senescence"

_nutrients, 2020, doi:10.3390/nu13010041_

Round 1
Reviewer 1 Report
The manuscript presents the study where the beneficial effects of creatine have been investigated against the liver damage associated with doxorubin treatment in an animal model. Even though the study was well designed and performed, the quality of presentation needs to be improved. If considered for revision, the manuscript needs a thorough language editing before consideration as a few errors are only pointed out as the minor comments.
1) Introduce doxorubicin in the abstract before the abbreviation. Similarly, introduce the full term before using the acronym for all the acronyms used (both in the abstract and in the text).
2) Line 24, "were shown" is not an appropriate usage which could be better replaced with "were observed" or "were noted" with DOX treatment but were diminished in Cr combined with DOX group.
3) Line 26: Why oxidative stress is noted listed among the others like "inflammation, fibrosis and senescence"?
4) Line 19-20, it would be ideal to mention clearly the methodologies followed in the study?
5) Line 32 is a bit confusing and it would be best to rephrase it for clarity as "such as breast, bladder and stomach cancer and leukemia"
6) "cell death [4] resulting in cyclical reaction which releases reactive oxygen species (ROS) [1,5]". What is "cyclical reaction" here? The cited references does not indicate any facts related to this statement?
7) "caused by DOX is just one way to cause" the use of the word "cause" latter in this phrase is sounding odd and can be replaced with any other suitable word. The word "cause" occurs 4 times between the sentences 40-42.
8) Although rare, severe cases of hepatotoxicity have also 48 been observed when multiple chemotherapeutic agents are used "together with DOX" [12]. I think that the authors wanted to convey the fact added with the phrase.
9) "would affect liver damage in rats and involved mechanism" unclear phrase
10) skeletal and cardio myocytes? What is the cell type of the former? (the lines in the same paragraph mentions skeletal muscle/skeletal muscle tissue?
11) Please expand the terms ALP, ALT, AST, etc., when occurring for the first time in the text.
12) Significance was set at p=0.05 or p≤0.05? Explain.
13) Liver to body "weight" ratio in Fig 1B legend
14) We found elevated liver-to-body weight ratio in CTRL/DOX when compared to reference range and CTRL/SAL. The labelling of the group (CTRL/DOX) is mismatching and hence creating confusion in the text than in the figure 1.
15) The legends of y-axis of many figures are small in font size and so looks unclear/blurry. The quality of few images were also low. The signs of significance should be indicated in bigger size to see it distinctly clear.
16) The legends of Fig 3 is incomplete in the sixth bar.
17) The figures, legends and description should be clear and understandable on its standalone manner without referring to its related text. For ex: %5-mC in Fig 6 will not be understood without referring to the text.
18) The scientific reasoning behind the observed results were not properly justified with proper explanation at many instance with relevance to cellular senescence, oxidative stress, inflammation and liver fibrosis. For example, how MCP is related to liver injury?
19) Moreover, is the gene detection for all MCP or specific for MCP-1?
20) Without statistical significance, the results of Figure 5 is more of a overstatement of the observed data and needs revision?
Minor comments:
1) Doxorubicin-induced - introduce hyphen where applicable
2) the potential therapeutic effect of Cr (not for Cr)
3) Line 18, the liver morphology
4) for indices of "liver" fibrosis
5) Line 21, The ALT
6) Line 23, supplemented with Cr
7) Why the words (Doxorubicin; Creatine) were in initial capitals than the others in the keywords?
8) Line 35: mechanisms underlying the anti-cancer effects ARE multifaceted in its cytotoxic effects AND ARE not fully understood
9) Line 42: "elevate endotoxin" and not "elevated".
10) Liver is one of the organs that affected by DOX. Language error in this sentence.
11) "how DOX could lead to liver damage" and not "leads"
12) Line 63: "no report has shown" is not an appropriate language. Moreover, the word "shown" has already been used in the same sentence and sounds odd.
13) Line 68: "cellular damage would be decreased"
14) Line 91: "each rat in the tubes. The tubes.."
Author Response
Reviewer1:
The manuscript presents the study where the beneficial effects of creatine have been investigated against the liver damage associated with doxorubicin treatment in an animal model. Even though the study was well designed and performed, the quality of presentation needs to be improved. If considered for revision, the manuscript needs a thorough language editing before consideration as a few errors are only pointed out as the minor comments.
Response: The authors appreciate all the reviewer’s comments and thank the reviewer for taking the time to review the manuscript. All the comments have been addressed as suggested by the reviewer.
- Introduce doxorubicin in the abstract before the abbreviation. Similarly, introduce the full term before using the acronym for all the acronyms used (both in the abstract and in the text).
Response: The authors have addressed this concern and have introduced the full term before using any acronyms in both the abstract and the full text.
- Line 24, "were shown" is not an appropriate usage which could be better replaced with "were observed" or "were noted" with DOX treatment but were diminished in Cr combined with DOX group.
Response: The authors apologize for this mistake. We have corrected the mistake in line 36 to 37.
3) Line 26: Why oxidative stress is noted listed among the others like "inflammation, fibrosis and senescence"?
Response: The authors apologize for this mistake. We have changes “The data suggest that Cr reduced DOX-induced hepatotoxicity by attenuating inflammation, fibrosis and senescence.” to “The data suggest that Cr protected against DOX-induced hepatotoxicity by attenuating fibrosis, inflammation, oxidative stress, and senescence.” (line 41-42)
4) Line 19-20, it would be ideal to mention clearly the methodologies followed in the study?
Response: Thank you for the great suggestions. The authors have changed the methodologies in the abstract to match the flow of the study. We have changed “liver morphology, collagen deposition, senescence, and inflammation were examined in liver tissue. Furthermore, mRNA levels for indices of fibrosis, inflammation, oxidative stress and senescence-related genes were measured.” to “markers of hepatoxicity were analyzed using liver-to-body weight ratio, aspartate transaminase (AST)-to- alanine aminotransferase (ALT) ratio, alkaline phosphatase (ALP), lipemia, and T-Bilirubin. In addition, hematoxylin and eosin (H&E) staining, Picro-Sirius Red staining, and immunofluorescence staining for CD45, 8-OHdG, and β-galactosidase were performed to evaluate liver morphologies, fibrosis, inflammation, oxidative stress, and cellular senescence, respectively.” in line 27 to 28.
5) Line 32 is a bit confusing and it would be best to rephrase it for clarity as "such as breast, bladder and stomach cancer and leukemia"
Response: Authors have changed [such as breast cancer, bladder cancer, stomach and leukemia] to [such as breast cancer, bladder cancer, stomach cancer and leukemia] in line 49.
6) "cell death [4] resulting in cyclical reaction which releases reactive oxygen species (ROS) [1,5]". What is "cyclical reaction" here? The cited references does not indicate any facts related to this statement?
Response: The authors apologize for the confusion. Authors have changed “cell death [4] resulting in cyclical reaction which releases reactive oxygen species (ROS) [1,5]” to “DOX undergoes redox cycling leading to the generation of reactive oxygen species (ROS)” in line 56.
7) "caused by DOX is just one way to cause" the use of the word "cause" latter in this phrase is sounding odd and can be replaced with any other suitable word. The word "cause" occurs 4 times between the sentences 40-42.
Response: Thank you for the suggestion, the authors have changed the word cause to other phrases. Specifically, we changed the text to “damage to cellular lipids, proteins, and DNA, which eventuates to cell death [6]. Doxorubicin treatment has also been found to promote systemic inflammation by damaging the small intestine and releasing microbial endotoxins into circulation thus stimulating pro-inflammatory pathways and enhancing hepatic inflammation” in line 55-58.
8) Although rare, severe cases of hepatotoxicity have also 48 been observed when multiple chemotherapeutic agents are used "together with DOX" [12]. I think that the authors wanted to convey the fact added with the phrase.
Response: Thank you for the suggestions. The authors have addressed this concern and have removed the sentence in line 72 to 73.
9) "would affect liver damage in rats and involved mechanism" unclear phrase
Response: We appreciate the comments. The authors have changed “would affect liver damage in rats and involved mechanism” to “explore mechanisms behind DOX-induced liver damage” in line 80.
10) skeletal and cardio myocytes? What is the cell type of the former? (the lines in the same paragraph mentions skeletal muscle/skeletal muscle tissue?
Response: We apologize for the confusion. The authors have changed added the cell types for skeletal tissue in the text. “The skeletal and cardio myocytes” has been changed to “The skeletal muscle fibers and cardiomyocytes” in line 89.
11) Please expand the terms ALP, ALT, AST, etc., when occurring for the first time in the text.
Response: We appreciate the comment. Authors have addressed this comment and used the full term before using the acronym.
12) Significance was set at p=0.05 or p≤0.05? Explain.
Response: Authors apologize for this mistake. We consider p value smaller than 0.05 as significant. We have changed this statement in methods session in line 203.
13) Liver to body "weight" ratio in Fig 1B legend
Response: Thank you for the comment. Authors have added the missing word in line 248 and added “weight” in the text as well.
14) We found elevated liver-to-body weight ratio in CTRL/DOX when compared to reference range and CTRL/SAL. The labelling of the group (CTRL/DOX) is mismatching and hence creating confusion in the text than in the figure 1.
Response: Authors apologize for this confusion. The reference line has been removed to avoid confusion. In term of the text format, we have updated the figure 1 label.
15) The legends of y-axis of many figures are small in font size and so looks unclear/blurry. The quality of few images were also low. The signs of significance should be indicated in bigger size to see it distinctly clear.
Response: we are sorry that the figures were not clear enough. All figures have been updated with bigger fonts for y axis and high quality of images. We have also updated the high-quality of figures in the system.
16) The legends of Fig 3 is incomplete in the sixth bar.
Response: Thank you for pointing that out. We have updated the picture. The older one was pasted to the word incorrectly.
17) The figures, legends and description should be clear and understandable on its standalone manner without referring to its related text. For ex: %5-mC in Fig 6 will not be understood without referring to the text.
Response: Authors appreciate and agree with the reviewer’s concern regarding the clarity of the legend and description of all figures. Specifically, authors have changed the figure legend of figure 6 to make sure it is clear without referring to the text.
18) The scientific reasoning behind the observed results were not properly justified with proper explanation at many instance with relevance to cellular senescence, oxidative stress, inflammation and liver fibrosis. For example, how MCP is related to liver injury?
Response: Thank you for the comments. The authors agree with the reviewer. More explanations have been added to justify the observed results (line 307-308). Monocyte chemoattractant protein-1 (MCP-1) expression was shown to be associated with cellular senescence. In hepatic liver injury, the secretion of MCP-1 will increase upon increased senescence [1].
19) Moreover, is the gene detection for all MCP or specific for MCP-1?
Response: We appreciate reviewer’s comments. Authors have evaluated MCP-1 specifically. We have fixed this in the revised manuscript.
20) Without statistical significance, the results of Figure 5 is more of a overstatement of the observed data and needs revision?
Response: Authors agree with the reviewer. We repeated the experiments for mRNA expression of MCP-1, with more samples being detected, we have shown statistically decreased in both Cr treatment when compared to CTRL/DOX. Please refer to new figure on page 14.
Minor comments:
1) Doxorubicin-induced - introduce hyphen where applicable
Response: Authors have added hyphen where it was needed.
2) the potential therapeutic effect of Cr (not for Cr)
Response: Authors have fixed this mistake in line 20.
3) Line 18, the liver morphology
Response: Authors have fixed this mistake in line 29.
4) for indices of "liver" fibrosis
Response: Authors have added the word “liver” to the revised manuscript in line 26.
5) Line 21, The ALT
Response: Authors have fixed this mistake in the revised manuscript in line 34.
6) Line 23, supplemented with Cr
Response: Authors have fixed this mistake in the revised manuscript in line 35.
7) Why the words (Doxorubicin; Creatine) were in initial capitals than the others in the keywords?
Response: Authors apologize for this mistake. We have fixed this mistake (line 43) in the revised manuscript and standardized all the naming style.
8) Line 35: mechanisms underlying the anti-cancer effects ARE multifaceted in its cytotoxic effects AND ARE not fully understood
Response: Authors have fixed this mistake in the revised manuscript in line 52.
9) Line 42: "elevate endotoxin" and not "elevated".
Response: Authors have fixed this mistake in the revised manuscript in line 60-61.
10) Liver is one of the organs that affected by DOX. Language error in this sentence.
Response: Authors have fixed this mistake in the revised manuscript in line 70.
11) "how DOX could lead to liver damage" and not "leads"
Response: Authors have fixed this mistake in the revised manuscript in line 76.
12) Line 63: "no report has shown" is not an appropriate language. Moreover, the word "shown" has already been used in the same sentence and sounds odd.
Response: Authors appreciate the comment. We have changed the phrasing in line 76.
13) Line 68: "cellular damage would be decreased"
Response: Authors have fixed this mistake in the revised manuscript.
14) Line 91: "each rat in the tubes. The tubes.."
Response: Authors have fixed this mistake in the revised manuscript in line 129 to 130.
- Lee, W.J.; Jo, S.Y.; Lee, M.H.; Won, C.H.; Lee, M.W.; Choi, J.H.; Chang, S.E. The Effect of MCP-1/CCR2 on the Proliferation and Senescence of Epidermal Constituent Cells in Solar Lentigo. Int J Mol Sci 2016, 17, doi:10.3390/ijms17060948.
Reviewer 2 Report
The study covers an interesting topic, but results are not fully convincing and sometimes confused, as it is not clear from the text if experiments were done in this manuscript or are referred to previous studies (see below). Besides, the potential side effects of Cr are not introduced at all and the translational potential of this study in other species is not treated.
A few points to fix
In the Introduction, at 54, when introducing creatine, please indicate what are the main biochemical/physiological function of this substance, besides stating that “Additionally, Cr functions as an antioxidant…”
The sentence “. We also hypothesized that senescence and cellular damage will be decreased when Cr supplementation is combined with DOX treatment.” is not an appropriate conclusion for an Introduction. What activities did authors carry out in the manuscript related to this hypothesis?
In the Results section, testing of liver health via serum chemistry. Could authors clearly explain what parameters were tested (and reported in (Supplementary table 2)?
At line 159, the sentence “We found elevated liver-to-body weight ratio in CTRL/DOX 159 when compared to reference range and CTRL/SAL [21].” Is unclear. Is this sentence referred to the manuscript or to a previous study?
The same can be applied to the sentence “Although the fibrotic difference between groups is not visible through naked eyes 176 (Figure 2A); when tissue sections were quantified with an established method [20], there was a 177 substantial increase in fibrosis in CTRL/DOX when compared to CTRL/SAL.”
There is no point in stating that “However, due to the large variance between samples, the difference was not statistically significant.” If the difference is not significant, this means that there is no difference, it is pointless to see that there is a difference but it not significant
Before the sentence “However, we found that the mRNA levels of a-SMA (data not shown) did not show a significant change among different groups, while there was a significant change between CTRL/DOX and 2% Cr/DOX in the mRNA expression of Fibronectin-1(Figure 2C).” authors should explain why these genes were tested and the methodology used
In the sentence “MCP is a widely used biomarker for cellular 235 senescence [28]”, it is unclear what MCP is. In the same sentence, it is not explained what senescence staining is
Reviewer 3 Report
One might interested in effects of Creatine on malondialdehyde and cytokines (e.g. IL-6 and ILβ)in Doxorubicin Induced Liver 2 Damage.
The authors should examine the efficacy of Creatine on oxidative stress and inflammation.
Round 2
Reviewer 1 Report
The manuscript have improved greatly in the quality of the presentation; however, up reading the manuscript correlating with the data, there were several flaws in the results (without statistics) and interpretation.
New comments:
Major:
1) While changed in lines 70-71, why the word "cardiomyocytes" was not changed at line 73?
2) skeletal myofibers, skeletal muscle myofibers and skeletal muscle are the three different terms used to describe the same detail between lines 69-74. Please use consistent terms to describe the same detail.
3) Line 73, while the order of words are different at other places, why it is mentioned as "cardiac myocytes and skeletal muscle" vice versa here? Please mention the facts consistently.
4) Line 74, it should be "the present study also aimed to evaluate.." as at line 61, the aim was described in past tense.
5) Line 242-244: "Cellular senescence is defined as cell cycle arrest and the incapability of cellular proliferation. These phenomena mainly occur due to genetic changes, yet more recently, it was identified that it could be induced by external factors such as radiation and drug treatments."
What does the phrase "these phenomena" refer to - "Cellular senescence" or "cell cycle arrest and the incapability of cellular proliferation"?
If referred to "Cellular senescence", the next sentence should be "This phenomenon mainly occurs due to..."
If referred to "cell cycle arrest and the incapability of cellular proliferation", the next sentence should be "These phenomena mainly occur due to genetic changes, yet more recently, it was identified that THEY could be.."
Please note the singular/plural - noun/pronouns used accordingly.
6) For my comment in the previous review,
"Without statistical significance, the results of Figure 5 is more of a overstatement of the observed data and needs revision?
Response: Authors agree with the reviewer. We repeated the experiments for mRNA expression of MCP-1, with more samples being detected, we have shown statistically decreased in both Cr treatment when compared to CTRL/DOX. Please refer to new figure on page 14.
the authors have responded which is still not convincing. There are few things that need to be addressed:
a) First of all, there is no change in Fig 5 before and after revision?
b) The authors, in the revision reply letter, have mentioned that "more samples" were detected, but still the indication "four biological replicates" are the same in the figure legend.
c) "Images were obtained under 20X magnification" should be moved at the end of (A) to correlate with the data of imaging
d) As compared to the MCP-1 expression of CTRL/DOX group, when the decrease in the expression of 2% Cr/DOX is having significance of ** with p<0.01, one would expect the 4%/2% Cr/DOX group having a lesser value than 2% Cr/DOX to be with significance of ** with p<0.01, and not * with p<0.05. Please verify the statistics and explain.
7) Figure 1:
a) Text: We found an elevated liver-to-body weight ratio in CTRL/DOX when compared to CTRL/SAL - this is an overstatement of the observation without any statistical analysis and resulting significance?
b) Text: Significant decrease in liver-to-body weight ratio was observed in 2% Cr/DOX when compared with CTRL/DOX group (p< 0.001) - even though it looks true, what about the significance of 4%/2% Cr/SAL group versus CTRL/DOX group, which would also be significant if analyzed statistically? If so, then it is the per se effect of 4%/2% Cr/SAL and not a protective effect of the supplementation from the damage induced by DOX in the CTRL/DOX
group?
8) Many of the results of 4%/2% Cr/SAL in Figure 3, 4, and 6 shows the per se effect of SAL seems to have some influence on the experimental parameters which were not considered for statistical analysis? Thus, this results in improper analysis and conclusion of the protective effects of supplementation disregarding the partial (or full?) contribution of the protective effects of SAL?
9) Above all, in Figure 3C, why no statistical interpretation was performed? The description of the results from this figure is ambiguous without statistics and not correct. Finally, the IL-1B induction of 4%/2% Cr/SAL and CTRL/DOX seems to be similar which indicates that the Cr supplementation or saline per se seems to induce inflammation? Even if its the case, while 4%/2% Cr/SAL per see is inducing IL-1B expression, how it could be protective in the presence of DOX? This brings lots of confusion to the whole experiment/model studied?
10) Ideally, at least, the statistics should be performed for control versus supplement, control versus disease-induced group, disease-induced group versus disease-induced plus supplement group? In addition, the comparison of solvent of supplement and disease-induced plus supplement group should be performed when applicable.
Minor:
1) Line 16: "to minimize for DOX-induced side effects" - remove "for" in this phrase
2) Line 65-66: "its use as a dietary supplement is generally considered safe and touted
for..."
3) Figure 1 legend: "Evaluation OF the effect..."
Author Response
The manuscript have improved greatly in the quality of the presentation; however, up reading the manuscript correlating with the data, there were several flaws in the results (without statistics) and interpretation.
New comments:
Major:
1) While changed in lines 70-71, why the word "cardiomyocytes" was not changed at line 73?
Response: Thank you for the comment. We have changed the terms to “cardiomyocytes”
2) skeletal myofibers, skeletal muscle myofibers and skeletal muscle are the three different terms used to describe the same detail between lines 69-74. Please use consistent terms to describe the same detail.
Response: I appreciate the comment. We have changed some terms to be consistent with “skeletal myofibers”
3) Line 73, while the order of words are different at other places, why it is mentioned as "cardiac myocytes and skeletal muscle" vice versa here? Please mention the facts consistently.
Response: Thank you for the comment. I have changed to order to make it consistent.
4) Line 74, it should be "the present study also aimed to evaluate.." as at line 61, the aim was described in past tense.
Response: Thank you for the comment. We have made the changes.
5) Line 242-244: "Cellular senescence is defined as cell cycle arrest and the incapability of cellular proliferation. These phenomena mainly occur due to genetic changes, yet more recently, it was identified that it could be induced by external factors such as radiation and drug treatments."
What does the phrase "these phenomena" refer to - "Cellular senescence" or "cell cycle arrest and the incapability of cellular proliferation"?
If referred to "Cellular senescence", the next sentence should be "This phenomenon mainly occurs due to..."
If referred to "cell cycle arrest and the incapability of cellular proliferation", the next sentence should be "These phenomena mainly occur due to genetic changes, yet more recently, it was identified that THEY could be.."
Please note the singular/plural - noun/pronouns used accordingly.
Response: Thank you for the comment. We have made the changes.
6) For my comment in the previous review,
"Without statistical significance, the results of Figure 5 is more of a overstatement of the observed data and needs revision?
Response: Authors agree with the reviewer. We repeated the experiments for mRNA expression of MCP-1, with more samples being detected, we have shown statistically decreased in both Cr treatment when compared to CTRL/DOX. Please refer to new figure on page 14.
the authors have responded which is still not convincing. There are few things that need to be addressed:
a) First of all, there is no change in Fig 5 before and after revision?
Response: Thank you for the comment. We are sorry for the confusion. I am not sure if you used the track of change version of a revised manuscript. Because it will still show both old and new version, with a red line on the old version. Now the previous track of change has been accepted. You should be able to see that new figure 5B is different now. And now the difference between CTRL/DOX and 2% Cr/DOX and CTRL/DOX and 4%/2% Cr/DOX are significant.
b) The authors, in the revision reply letter, have mentioned that "more samples" were detected, but still the indication "four biological replicates" are the same in the figure legend.
Response: Thank you for the comment. We are sorry for this mistake. We added two new sample sizes, and now we have changed to six biological replicates.
- c) "Images were obtained under 20X magnification" should be moved at the end of (A) to correlate with the data of imaging
Response: Thank you for the comment. We have moved the phrase to the place suggested by the reviewer.
d) As compared to the MCP-1 expression of CTRL/DOX group, when the decrease in the expression of 2% Cr/DOX is having significance of ** with p<0.01, one would expect the 4%/2% Cr/DOX group having a lesser value than 2% Cr/DOX to be with significance of ** with p<0.01, and not * with p<0.05. Please verify the statistics and explain.
Response: Thank you for the comment. We have verified the statistics. The new figure 5 showed **p<0.01 in both 2% Cr/DOX and 4%/2% Cr/DOX when compared to CTRL/DOX. And *p<0.05 when CTRL/DOX compared to CTRL/SAL.
7) Figure 1:
a) Text: We found an elevated liver-to-body weight ratio in CTRL/DOX when compared to CTRL/SAL - this is an overstatement of the observation without any statistical analysis and resulting significance?
Response: Thank you for the comment. The statistical difference between CTRL/DOX and CTRL/SAL is not significant. So we have changed the text to “We found a trend of …”
b) Text: Significant decrease in liver-to-body weight ratio was observed in 2% Cr/DOX when compared with CTRL/DOX group (p< 0.001) - even though it looks true, what about the significance of 4%/2% Cr/SAL group versus CTRL/DOX group, which would also be significant if analyzed statistically? If so, then it is the per se effect of 4%/2% Cr/SAL and not a protective effect of the supplementation from the damage induced by DOX in the CTRL/DOX
group?
Response: Thank you for the comment. We have checked the statistics again, the liver to body weight ratio in 4%/2% Cr/SAL significantly decreased when compared to CTRL/DOX (we have added it to the new figure 1B). When we compared both 4%/2% Cr/SAL and CTRL/DOX to the CTRL/SAL respectively, we can conclude that the difference between CTRL/DOX and 4%/2% Cr/SAL indicates feeding with 4%/2% Cr causes the opposite effect as which the DOX caused to the liver. However, this is not affecting us to conclude the rescuing trend of liver to body weight ratio proven by close to the control in 2% Cr/DOX and 4%/2% Cr/DOX group. To further explore the potential stress caused in 4%/2% Cr/SAL group, more studies are needed. We have noticed the side effects and put them in the discussion.
8) Many of the results of 4%/2% Cr/SAL in Figure 3, 4, and 6 shows the per se effect of SAL seems to have some influence on the experimental parameters which were not considered for statistical analysis? Thus, this results in improper analysis and conclusion of the protective effects of supplementation disregarding the partial (or full?) contribution of the protective effects of SAL?
Response: It is a great question. The increase of expression shown on NF-kB gene expression and IL-1 expression, and NRF-2 expression are significant when compared to CTRL/SAL. We did not find significance of global methylation in 4%/2% Cr/SAL when compared to CTRL/SAL. We have modified the text based on the new figures. As stated in the text in both results and the discussion, because we did not see positive CD-45 staining and 8-OHdG staining, we did not further evaluate the potential damage of 4%/2% Cr in control group. However, these results would suggest a potential inflammatory and oxidative stress caused by 4%/2% Cr in healthy rats. We have pointed out that “4%/2% Cr/SAL presented a moderate increase mRNA expression of the liver oxidative stress biomarker NRF-2 and the inflammation biomarkers NF-kB and IL-1 b along with observable lymphocytic penetration…. The higher dose of Cr, however, could potentially increase inflammation and oxidative stress in the healthy rat liver.” We have put it in our discussion (third paragraph of the discussion)
9) Above all, in Figure 3C, why no statistical interpretation was performed? The description of the results from this figure is ambiguous without statistics and not correct. Finally, the IL-1B induction of 4%/2% Cr/SAL and CTRL/DOX seems to be similar which indicates that the Cr supplementation or saline per se seems to induce inflammation? Even if its the case, while 4%/2% Cr/SAL per see is inducing IL-1B expression, how it could be protective in the presence of DOX? This brings lots of confusion to the whole experiment/model studied?
Response: Sorry for the confusion. We have rerun the statistical analysis between CTRL/SAL vs. Cr/SAL, CTRL/SAL vs. CTRL/DOX, and CTRL/DOX vs. Cr/DOX. We have found both increased expression of IL1-beta in 4%/2% Cr/SAL and CTRL/DOX when compared to CTRL/SAL. We have put these changes on figure3. We apologize for the confusion. Although the 4%/2% Cr/SAL group showed increased expression of IL-1beta, this was not observed in the CD-45 staining. On the other hand, we observed decreased damage of 4%/2% Cr/DOX when compared to CTRL/DOX. So, our data showed a protective effect when 4%/2% Cr supplemented with DOX but not the CTRL group.
10) Ideally, at least, the statistics should be performed for control versus supplement, control versus disease-induced group, disease-induced group versus disease-induced plus supplement group? In addition, the comparison of solvent of supplement and disease-induced plus supplement group should be performed when applicable.
Response: Thank you for the comment. We have repeated the statistical analysis. And ran the analysis between CTRL/SAL vs. Cr/SAL, CTRL/SAL vs CTRL/DOX, and CTRL/DOX vs Cr/DOX. We have added the modified change to all figures. Specifically, we added
- ***p<0.001 of 4%/2% Cr/SAL vs CTRL/DOX to figure 1B upper pannel.
- *p<0.05 of 4%/2% Cr/SAL vs CTRL/SAL to figure 3B.
- *p<0.05 of 4%/2% Cr/SAL and CTRL/DOX vs CTRL/SAL to figure 3C,
- *p<0.01 of 4%/2% Cr/SAL vs CTRL/SAL to figure 4B
- *P<0.05 of CTRL/DOX vs CTRL/SAL to figure 5B
- *P<0.05 of 2% Cr/DOX and 4%/2% Cr/DOX vs CTRL/SAL to figure 5B
We did not find any other significant between groups in all figures if not indicated.
Minor:
1) Line 16: "to minimize for DOX-induced side effects" - remove "for" in this phrase
Response: Thank you for the comment. We have removed the word.
2) Line 65-66: "its use as a dietary supplement is generally considered safe and touted
for..."
Response: Thank you for the comment. We have changed the sentence according to the reviewer’s comment.
3) Figure 1 legend: "Evaluation OF the effect..."
Response: Thank you for the comment. We have added the work to the figure legend.

Reviewer 2 Report
The authors have provided an exhaustive point-by-point reply to all concerns raised, improving the clarity of their manuscript
Author Response
We thank you for the reviewer's comments.